# A Comparative Study of Cellulose Ethers as Thermotropic Materials for Self-Tracking Solar Concentrators

**DOI:** 10.3390/molecules27238464

**Published:** 2022-12-02

**Authors:** Francesco Galeotti, Lorenzo Scatena, Franco Trespidi, Mariacecilia Pasini

**Affiliations:** 1Istituto di Scienze e Tecnologie Chimiche “G. Natta” (SCITEC), Consiglio Nazionale delle Ricerche, Via A. Corti 12, 20133 Milano, Italy; 2Ricerca sul Sistema Energetico (RSE), Strada Torre della Razza, Loc. Le Mose, 29122 Piacenza, Italy

**Keywords:** cellulose derivatives, hydrogels, thermotropic materials, light trapping, photovoltaic windows, smart windows, solar concentrator, renewable energy

## Abstract

The continuous growth in energy demand requires researchers to find new solutions to enlarge and diversify the possible ways of exploiting renewable energy sources. Our idea is the development of a solar concentrator based on trapping the luminous radiation with a smart window. This system is able to direct light towards the photovoltaic cells placed on window borders and produce electricity, without any movable part and without changing its transparency. Herein, we report a detailed study of cellulose ethers, a class of materials of natural origin capable of changing their state, from transparent aqueous solution to scattering hydrogel, in response to a temperature change. Cellulose thermotropism can be used to produce a scattering spot in a window filled with the thermotropic fluid to create a new kind of self-tracking solar concentrator. We demonstrate that the properties of the thermotropic fluid can be finely tuned by selecting the cellulose functionalization, the co-dissolved salt, and by regulating their dosage. Lastly, the results of our investigation are tested in a proof-of-concept demonstration of solar concentration achieved by thermotropism-based light trapping.

## 1. Introduction

Thermotropic polymers are a class of materials able to switch their state, from clear to strongly scattering, in response to a temperature change. Thanks to the reversibility of this transparent/opaque transition, they are attractive for photonic applications, particularly in smart windows, where they can play a critical role in enhancing the energy efficiency and the comfort level of indoor spaces [1].

The physical mechanism under thermotropism relies on a phase transition, occurring at the critical temperature, from polymer homogeneously dissolved in the solvent to the appearance of partially undissolved/aggregated polymer chains. Below the phase transition temperature, that in case of polymer gels or blends is often called “lower critical solution temperature” (LCST), the refractive indices of the polymer and the solvent are almost identical, so that the system exhibits a transparent state. When the temperature rises above the LCST, the refractive index of the aggregated polymer phase increases, generating an index mismatch with respect to the matrix, which causes light scattering [2].

In addition to reflecting back part of the light passing through the material and therefore acting similarly to an automatic temperature-driven light protection, the scattering state of a thermotropic polymer opportunely confined in a transparent window can be exploited to trap light into waveguide modes. In fact, as light is backscattered by the thermotropic polymer in every direction, part of it will hit the outer surfaces of the cell windows with angles satisfying the conditions for total internal reflection. Consequently, light will be trapped inside the window.

In our idea, sunlight passing through a smart window is conveyed by waveguide to the window edges, where small photovoltaic cells are positioned. The final goal is to feed the solar cells with more radiation than what they would normally capture, hence boosting electricity conversion.

Several efforts have been made to maximize the efficiency of both organic and inorganic solar cells [3,4], and these include the development of solar concentrators based on different mechanisms such as luminescent solar concentrators [5,6], that facilitate the conversion of the wavelength of absorbed light into the zone of maximum efficiency of the cell, or anti-reflective coatings [7,8,9], that allow for the minimization of the energy loss due to light reflection by the external surface of solar cells. In this perspective, we propose thermotropic cellulose ethers as possible materials enabling a new type of solar concentrator.

In our system, depicted in Figure 1, to capture the solar radiation in the smart window by waveguide and let it reach the solar cells without further scattering phenomena, the material filling the window should maintain its transparency except for a small area where sunlight is concentrated. To do so, sunlight must be concentrated on a tiny spot by a lens, so that the LCST of the thermotropic material is reached only in this confined area, while the rest of the window remains in its original transparent state. The transition from transparent to scattering state is triggered by the IR bands of solar radiation. Water, which is the main component of the thermotropic fluid, is in fact able to adsorb part of this radiation, generating the local heating that starts the transition. The reversibility of the process assures that, as the direction of sun rays reaching the lens changes during the day, the position of scattering spot on the window continuously moves (Figure 1). Because this device does not require a mechanical sun-tracking system, we can describe it as self-tracking solar concentrator [10]. These kinds of photovoltaic windows have the advantage of considerably reducing the extensions of solar cells with respect to photovoltaic roofs, and thus lowering their cost, while preserving the original window function and appearance. This system can also become a design element that enables the elimination of the entry of direct light in the internal ambient, creating a more pleasant and soft light. Even though a comprehensive analysis of possible technological solutions to solar concentration is out of the scope of this article, it is worth to remark here that, despite the low efficiency generally expected for the self-tracking sun concentration, this approach provides great advantages in terms of cost and integration with residential buildings, with respect to the more traditional mechanical tracking. Therefore, any advancement of knowledge in this field could be essential for the development of future environmentally friendly energy production systems.

Chemically, thermoresponsive polymers have both hydrophilic and hydrophobic subunits. The hydrophilic subunits can form hydrogen bonds with water and keep the polymer chains in a random coil-shaped, hydrated state. Thus, the polymer is dissolved in water leading to a single, homogenous, transparent phase. When the temperature increases beyond LCST, the conformation is changed from coil to globule form. The hydrophilic subunits become inaccessible to water molecules, causing the dehydration of polymer chains and, consequently, the formation of a biphasic, nonhomogeneous, scattering system. Given the requirement of hydrophilic and hydrophobic domains on their chain, LCST-type thermotropic polymers can belong to different chemical classes: ethers, alcohols, amides and polypeptides [11].

Among thermotropic materials, one of the most widely studied is poly(N-isopropylacrylamide), which forms thick hydrogels with LCST around 30–32 °C in its homopolymer form [12,13,14], and whose properties can be adapted and modified by mixing it with colloidal particles [15], with nanomaterials [16], or by copolymerization with different monomers [17,18]. Other examples of thermoregulated reversible sol/gel transition have been reported for different copolymers, e.g., ulvan-grafted poly(N-vinylcaprolactam) [19], poly(ethylene oxide-b-propylene oxide-b-ethylene oxide) (PEOPPO-PEO) [20], and poly(ethylene glycol-b-L-lactic acid-b-ethylene glycol) (PEG-PLLA-PEG) [21]. All these thermoresponsive matrices are characterized by LCST values close to physiological temperature, therefore they were specifically thought for the development of drug delivery systems.

For our application, the scattering medium should be chemically simple and economically affordable, with transition temperature within the range of 45–60 °C, high enough to be not reachable in standard sun conditions, and easily reachable by little sun concentration. For these reasons, a more suitable class of material to be considered for the scattering medium is represented by cellulose ethers.

Cellulose is a natural polymer characterized by a high hydrophilicity on its chain structure. Because cellulose forms strong intermolecular hydrogen bonds, however, it is insoluble in water. When a certain fraction of hydroxyl groups is substituted by hydrophobic groups such as methoxide groups, intermolecular hydrogen bonds are weakened, resulting in water solubility [22]. The resultant ether derivatives are called hydrophobically modified cellulose or water-soluble cellulose. However, the degree of substitution is crucial to achieve water-soluble cellulose, because too many hydrophobic groups make cellulose derivatives water-insoluble again. In the case of methylcellulose, typically the average degree of substitution of methoxy groups that provide solubility is 1.7–2.1 (out of 3 -OH groups) per anhydroglucose unit [23].

Cellulose ethers, like native cellulose, are not digestible, not toxic, not allergenic, and they are extensively used as a thickener and emulsifier in various food and cosmetic products, in laxative drugs and in the manufacturing of drug capsules [24]. More recently, hydrogels based on cellulose ethers have been proposed for more innovative applications like drug delivery [25,26], tissue engineering [27,28], and for smart windows [29,30,31].

Their easy availability, directly connected to the extensive industrial usage, their LCST generally around 40 °C and above, and the possibility of different chemical substitution leading to tunable optical properties, make cellulose ethers the perfect candidate materials for our scattering-based solar concentrator, with respect to other synthetic and natural polymers with thermotropic properties. Furthermore, this material perfectly meets the requirements of sustainability such as low cost, easy availability, abundance, non-toxicity and does not present disposal problems, creating a virtuous combination of energy and sustainability. Even though the concept of a self-tracking solar concentrator based on a scattering medium was proposed a few years ago, to our knowledge, no practical demonstration/proof-of-concept has ever been published [32]. Following this unexplored but innovative idea, we present here a study aimed at selecting the better matrix, based on cellulose ethers, to realize it and at evaluating its practical feasibility.

## 2. Results

For our study, we initially considered six different cellulose derivatives that can be easily found on the market. In this series, cellulose ethers are characterized by either different viscosity while sharing the same kind of substitution (methyl celluloses) or by different substituting groups (methyl, hydroxyethyl, hydroxypropyl or (hydroxypropyl)methyl), as summarized in Table 1.

In a preliminary screening aimed at testing the scattering capability of the selected materials, the diluted aqueous solutions were gradually heated with a hair-drier until we observed the formation of a scattering phase. The cuvette containing 1 wt % cellulose ether solutions quickly became opaque. By repeating the experiment in partially filled vials, we could verify by turning them upside-down that a liquid-gel transition accompanies the observed optical transition, increasing the viscosity of the mixture. By immersing the cuvette/vial in cold water, the temperature was quickly decreased, and the reversibility of the process could be visually confirmed (Figure 1).

The visual comparison between the different cellulose derivatives before and after reaching the LCST revealed some differences. The scattering phase of methyl-celluloses sill maintained a minimum of transparency which allowed to glimpse the text underneath. HEC did not show any evident scattering phase upon heating; therefore, it was not considered anymore in this study. HPC produced a narrow scattering solid immersed in the transparent matrix. HPMC scattering phase was highly opaque and perfectly hid the text behind (Figure 2). In general, the opacity occurs instantaneously as the transition temperature is reached and, as soon as it is cooled, the solution becomes transparent again.

Following this preliminary assay, the next step was to determine the LCST of the different cellulose derivatives. To do so, we recorded the variation in light transmission through the cuvette filled with the cellulose solution by the microscope camera while varying their temperature from 30 to 95 °C. After reaching the highest temperature, the temperature scan was repeated back until it reached the initial cold point.

As reported by the plot in Figure 3, all the tested solutions reached the LCST between 50 and 70 °C upon heating. As already noticed by eye observation, the hot-stage measurements confirmed that the transparency loss of the scattering state is different for the materials; the methyl-celluloses preserve a certain amount of transparency, while for HPC and HPMC, the transparency loss is more pronounced. This indicates that the chemical functionalization is a critical value affecting the scattering state of cellulose ethers. Specifically, the introduction of aliphatic chains with hydroxy functionalities produces a more efficient scattering phase with respect to substitution with methyl groups only. On the other hand, the scattering state of HPC resembles more a solid polymer than a hydrogel (see the corresponding photograph in Figure 2), indicating that hydroxypropyl functionalization leads to a highly compact globule phase. It must be considered here that the degree of substitution (the average number of substituent groups attached to the ring hydroxyls), which for commercial cellulose ethers is generally kept in the range 1.5–1.9 to assure the maximum water solubility, can also play a role in the scattering capacity. Upon cooling, the reverse process is more gradual for all the materials with respect to the heating one, resulting in slightly lower transition temperatures from scattering to transparent with respect to the forward LCST.

This screening provided a first indication that cellulose ethers can be suitable for developing a self-tracking solar concentrator based on waveguiding, provided that the proper polymer functionalization and thus the right LCST is selected. However, the kind of functionalization is not the only parameter that must be taken into consideration. In fact, salts are known to influence the temperature-induced phase transitions in aqueous solutions of thermosensitive polymers [33]. This is a key point especially when the process of gelification, swelling and dissolution and of cellulose ethers in the presence of bio-fluids (containing salts) are explored for drug-delivery purposes [34]. In general, salts may either enhance or reduce the hydrophobicity of a solute in water. The so-called “Hofmeister series” is an order of ions ranked in terms of how strongly they affect the hydrophobicity [35]. Because gelation of cellulose ethers is due the aggregation of hydrophobic groups when water becomes a poorer solvent for them due to temperature rise, the presence of ions with their resultant effect is capable of influencing this process.

For anions, a typical Hofmeister order is:SO_4_^2−^ > F^−^ > Cl^−^ > Br^−^ > NO_3_^−^ > ClO_4_^−^ > I^−^ > SCN^−^,
where ions on the left-hand side exhibit strong interactions with water molecules and, as a result, they tend to cause “salt-out” or to enhance hydrophobicity of a solute in water. The expected effect on cellulose ethers is to lower the LCST. In contrast, ions on the right-hand side are able to cause “salt-in”, which increases the solubility of a nonpolar solute, thus raising the LCST of the co-dissolved cellulose.

In this scenario, we evaluated the influence of two “salting-in” anions, I^−^ and SCN^−^, and one “salting-out” anion, namely Cl^−^, on the LCST of cellulose ethers. To better elucidate the effect of the anions, the counterion was always K^+^, and salt concentration was fixed at 0.5 M. As expected, in the presence of KI and KSCN all the cellulose ethers increased their LCST of 5–15 °C, while the addition of KCl lowered the LCST of the corresponding polymer of 5–10 °C (Figure 4). Moreover, as summarized by the plot in Figure 4d, MC-B, MC-C and HPC perfectly fulfilled the Hofmeister order, showing decreasing LCST in the order I^−^, SCN^−^, no salt and Cl^−^. MC-A and HPMC showed slightly higher values of LCST in the presence of I^−^ than with SCN^−^.

On the basis of the observations and data reported so far, we selected HPMC as the material with the better opacity and homogeneity of the scattering phase. In addition, we selected KCl as the co-dissolved salt because it could guarantee to keep the transition temperature of HPMC between 50 and 60 °C, a temperature range suitable for our application. However, a proof-of-concept test with a home-made glass window filled with the polymer-salt mixture evidenced the issue of insufficient viscosity of the mixture. In fact, the gel phase formed in correspondence of the concentrated light spot, tended to move around inside the fluid. This fact is detrimental for the envisaged application as self-tracking solar concentrator because the migration of the scattering spot away from the initial position lets the solar light pass through the window instead of being scattered and then trapped by waveguiding. To overcome this problem, we tested two new polymers with higher molecular weight, 86 and 120 kDa, respectively, and sufficient viscosity to avoid any migration effect. The main characteristics of the selected high viscosity materials and of HPMC 10, the one used in the preliminary assessment tests, are reported in Table 2.

In the plots showed in Figure 5, we report the scattering transition properties of the different HPMC derivatives in the presence of different concentrations of KCl. At this stage, the temperature cycle was reported in sequential mode (from T at time 0 to T at the end of the cycle) in order to better visualize the small differences imparted by different salt concentrations.

All the three HPMC batches showed similar temperature-dependent transparency variations, as evidenced by the shape of the plots in Figure 5. The similar values recorded in the flat bottom part of the curves indicate that the transparency loss after gelification was only slightly affected by viscosity. As expected, by increasing the salt concentration, the “salting-out” effected is augmented for all the polymer solutions, causing the lowering of the respective LCST. This concentration-dependency effect is more pronounced for HPMC10 and HPMC86, for which we observed a variation of 20 and 15 °C, respectively, between the solution with no salt and the one spiked with KCl 1.0 M. For HPMC120, the observed LCST variation is in the range of 5 °C, which suggests that the high molecular weight polymer is only partially affected by the presence of co-dissolved salts.

These last measurements allowed us to assess that the LCTS of HPMC con be finely tuned by adjusting the amount of KCl, which was a crucial requirement for testing the thermotropic fluid in a prototype cell for solar concentration.

For the proof-of-concept demonstration of self-tracking solar concentrator, we selected as thermotropic fluid 1 wt % aqueous solution of HPMC86 with 1.0 M KCl. In fact, this mixture fulfilled both the main requirements for this test: LCST around 45 °C and viscosity high enough to avoid the migration of the scattering mass inside the window.

This test was conducted using a cell with glass windows and internal dimensions of 160 × 160 × 4 mm, filled with the thermotropic fluid. We performed two different measurements on the same cell, by positioning the spot of the sunlight concentrated by a lens in the center of the cell and at 3 cm of distance from one side. The obtained solar concentration factor (CS) data are reported in Figure 6.

The CS values were obtained by measuring with an integration sphere the light reaching the lateral side of the cell. For each wavelength λ, the ratio between the value measured on the cell side (trapped light) and the one measured by orienting the integration sphere towards the sun (solar light) provided the concentration values $C\left(\lambda \right)$. The concentrated irradiance was then obtained according to the following equation:$$\mathrm{Irc}=\int_{350nm}^{1000nm}C\left(\lambda \right)\;Ref\left(\lambda \right)d\lambda$$
where Irc is the concentrated irradiance, $C\left(\lambda \right)\;$ is the concentration factor and $Ref\left(\lambda \right)$ is the reference solar spectrum AM 1.5 D [36,37]. The ratio between Irc and the reference solar irradiance Ir1s (1 sun), provided the CS values reported in the plot of Figure 6:CS = Irc/Ir1s

The average CS values in the investigated wavelength range were equal to 0.49 and 0.32 for the measurements at 3 cm and at 8 cm, respectively. These results indicate that the system in this non-optimized setup failed to concentrate the solar radiation. In fact, for CS values smaller than 1, the concentrated light reaching the window’s side is less than the non-concentrated light reaching the same area. Still, the amount of light trapped by our system and potentially usable to generate electricity is relevant, and the experiment demonstrated for the first time the potentiality of using thermotropic cellulose derivatives in self-tracking solar concentrators. The setup however demonstrated that a part of the light collected by the lens was deviated from its original direction through this optical device and in principle such behavior should happen for a certain range of incidence angles without the need of any mechanical tracking system.

To achieve a real working system, the main parameters must be optimized. Specifically, by regulating the thickness of the thermotropic fluid (internal distance between the two glass layers of the cell), the size and efficiency of the scattering spot and the performance of the focusing optics, it would be possible, in principle, to maximize the amount of light reaching the window edges and to reach the goal of CS values exceeding unity.

## 3. Materials and Methods

Cellulose ethers were purchased from TCI Europe N.V. (Brussels, Belgium) except for (hydroxypropyl)methyl cellulose batches, that were purchased from Merck KGaA (Darmstadt, Germany).

The transparency variation plots were obtained by using a Nikon Eclipse Te2000 (Tokyo, Japan) inverted microscope equipped with a Linkam hot-stage system (Redhill, UK). Micrographs were taken every 2 s while the sample temperature was varied between 30 to 95 °C and then back to the 30 °C, at the rate of 1 °C × s^−1^. The maximum temperature was maintained for 10 s before starting the return run. The same values of light intensity and exposition were used for all the experiments. The transparency data were extrapolated by measuring the average brightness of each micrograph with ImageJ [38]. Since the digital images are stored in the form of a matrix of numbers representing the brightness of each pixel, their average values go from 0 (total black, no transparency) to 255 (total white, complete transparency).

The solar concentration experiments were performed by focalizing the solar radiation on a glass cell of 160 × 160 × 4 mm filled with the thermotropic fluid, by using a PMMA Fresnel lens. The temperature of glass surface in correspondence of the concentrated light spot during a sunny day reached 50–55 °C. Data were acquired by positioning an integration sphere on the edge of the window and on the window’s face, towards the sun (Figure 7).

## 4. Conclusions

In this study, we proposed to use the phenomenon of thermotropism to produce a scattering spot in a window filled with a thermotropic fluid, to create a new kind of self-tracking solar concentrator. By investigating different cellulose ethers in aqueous solution, we demonstrated that the properties of the thermotropic fluid can be finely tuned by selecting the cellulose functionalization, the co-dissolved salt, and by regulating their dosage. Then, we tested the results of our study in a proof-of-concept demonstration of solar concentration, showing that a part of the light collected by the lens is effectively trapped and deviated from its original direction by thermotropic effect without the need of any mechanical tracking, even though the system needs some optimization.

It is worth noting that the perfectly sustainable thermotropic system developed here is highly tunable to respond to a certain temperature range; specifically, fluids with different LCST can be used in different geographical areas or can be easily replaced in the same window depending on the season, thanks to the low cost, low toxicity and easy availability of the chosen approach. A further advantage is that this kind of transparent photovoltaic window utilizes only the direct sun radiation, while the diffused skylight arising from the scattering of the direct solar beam by molecules or particulates in the atmosphere, which represents normally 10–15% of the total radiation, will be still available to illuminate the internal ambient. More importantly, conversely to what happens with luminescent solar concentrators, here the natural illumination is maintained without any chromatic modification. Therefore, this system, based only on a lens protruding from a transparent window, can possibly become a design element that, by preventing the entry of direct light, is able to create a more pleasant and soft illuminated ambient.

In conclusion, after proper optimization, the solution proposed could be a viable alternative to enlarge and diversify the possible ways of exploiting renewable energy sources.

## Data Availability

Not applicable.

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
