# Peer review of "A Comparative Study of Cellulose Ethers as Thermotropic Materials for Self-Tracking Solar Concentrators"

_molecules, 2022, doi:10.3390/molecules27238464_

Round 1

Reviewer 1 Report

The manuscript studied the performance of cellulose ethers as thermotropic materials for self-tracking solar concentrators. This study found that cellulose ethers were filled in a window and could be used to create a solar concentrator. The manuscript must be revised for publishing on Molecules considering the following remarks:

1. In introduction, the author should introduce the structure characteristics for becoming the thermotropic materials.

2. In 144-151, the author should explain the critical factors for affecting the scattering state in the six different cellulose derivatives.

3. How about the effect of the viscosity of cellulose ethers for the scattering state.

4. How about the effect of the sun light time for the structure of cellulose ethers and the visual change of the solar concentrator.

5. How about the effect of the concentration of cellulose ethers for the scattering state.

6. In lines 241-245, Irc was calculated by the equation. Did the method has the reference.

7. the author should present the conclusion.

8. The English and the tense of the manuscript should be improved. 

Reviewer 2 Report

This manuscript deals with a study on cellulose ethers as thermotropic materials.

The authors conducted a lot of experiments that depict the mechanism of thermotropic behaviors of various cellulose derivatives, and the proof-of-concept solar concentration performance was also evaluated. The study is interesting and well organized.

Some additional questions.

1) Why the authors used cellulose derivatives for the study. There are similar kind of bio-polymers which have thermotropic properties such as chitosan, chitin, and so on. 
2) Can we idetify the mechanism of thermotropic property of celluloses in this system using other parameter such as solubility parameter, so on? 
3) For demonstration of solar concentration, the viscosity seems quite important to prepare sample. Please explain. 

Reviewer 3 Report

The studies carried out will contribute to the knowledge of the materials tested and to the further development of environmentally friendly technologies. The structure and structure of the article is neat, logical and easy to follow; the published results are presented and analysed objectively; the figures are well-developed and informative. The article language is understandable, traceable. However, I would suggest that some shortcomings in the editing should be remedied.

To improve the quality of the article, I have the following specific suggestions:

Describe in more detail how the transparency was measured, what was the measurement setup, what metric and scale was used? What does .a.u. stand for in the diagram?

For each function in Figure 5, record the quantities on the axes

I really miss the conclusion chapter, the summary. What applications do you see for optimization and practical use.

Round 2

Reviewer 1 Report

1.  the tense of the manuscript should be improved.

Reviewer 3 Report

Authors have made the requested corrections.